# BatchBALD: Efficient and Diverse Batch Acquisition for Deep Bayesian Active Learning

**Andreas Kirsch**[*]      **Joost van Amersfoort**[*]      **Yarin Gal**

OATML
Department of Computer Science
University of Oxford
{andreas.kirsch, joost.van.amersfoort, yarin}@cs.ox.ac.uk

## Abstract

We develop BatchBALD, a tractable approximation to the mutual information between a batch of points and model parameters, which we use as an acquisition function to select multiple informative points jointly for the task of deep Bayesian active learning. BatchBALD is a greedy linear-time $1 - \frac{1}{e}$-approximate algorithm amenable to dynamic programming and efficient caching. We compare BatchBALD to the commonly used approach for batch data acquisition and find that the current approach acquires similar and redundant points, sometimes performing worse than randomly acquiring data. We finish by showing that, using BatchBALD to consider dependencies within an acquisition batch, we achieve new state of the art performance on standard benchmarks, providing substantial data efficiency improvements in batch acquisition.

## 1   Introduction

A key problem in deep learning is data efficiency. While excellent performance can be obtained with modern tools, these are often data-hungry, rendering the deployment of deep learning in the real-world challenging for many tasks. Active learning (AL) [7] is a powerful technique for attaining data efficiency. Instead of a-priori collecting and labelling a large dataset, which often comes at a significant expense, in AL we iteratively acquire labels from an expert only for the most informative data points from a pool of available unlabelled data. After each acquisition step, the newly labelled points are added to the training set, and the model is retrained. This process is repeated until a suitable level of accuracy is achieved. The goal of AL is to minimise the amount of data that needs to be labelled. AL has already made real-world impact in manufacturing [34], robotics [5], recommender systems [1], medical imaging [18], and NLP [31], motivating the need for pushing AL even further.

In AL, the informativeness of new points is assessed by an *acquisition function*. There are a number of intuitive choices, such as model uncertainty and mutual information, and, in this paper, we focus on BALD [19], which has proven itself in the context of deep learning [13, 30, 20]. BALD is based on mutual information and scores points based on how well their label would inform us about the true model parameter distribution. In deep learning models [16, 32], we generally treat the parameters as point estimates instead of distributions. However, Bayesian neural networks have become a powerful alternative to traditional neural networks and do provide a distribution over their parameters. Improvements in approximate inference [4, 12] have enabled their usage for high dimensional data such as images and in conjunction with BALD for Bayesian AL of images [13].

In practical AL applications, instead of single data points, batches of data points are acquired during each acquisition step to reduce the number of times the model is retrained and expert-time is

---

[*]joint first authors

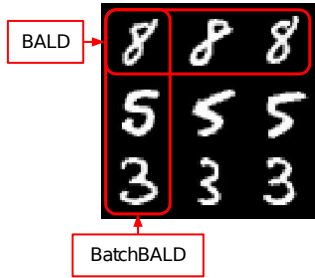

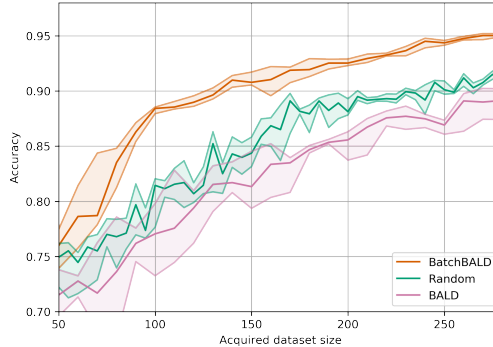

Figure 1: *Idealised acquisitions of BALD and BatchBALD*. If a dataset were to contain many (near) replicas for each data point, then BALD would select all replicas of a single informative data point at the expense of other informative data points, wasting data efficiency.

Figure 2: *Performance on* Repeated MNIST *with acquisition size 10*. See section 4.1 for further details. BatchBALD outperforms BALD while BALD performs worse than random acquisition due to the replications in the dataset.

requested. Model retraining becomes a computational bottleneck for larger models while expert time is expensive: consider, for example, the effort that goes into commissioning a medical specialist to label a single MRI scan, then waiting until the model is retrained, and then commissioning a new medical specialist to label the next MRI scan, and the extra amount of time this takes.

In Gal et al. [13], *batch acquisition*, i.e. the acquisition of multiple points, takes the top $b$ points with the highest BALD acquisition score. This naive approach leads to acquiring points that are individually very informative, but not necessarily so jointly. See figure 1 for such a batch acquisition of BALD in which it performs poorly whereas scoring points jointly ("BatchBALD") can find *batches* of informative data points. Figure 2 shows how a dataset consisting of repeated MNIST digits (with added Gaussian noise) leads BALD to perform worse than random acquisition while BatchBALD sustains good performance.

Naively finding the best batch to acquire requires enumerating all possible subsets within the available data, which is intractable as the number of potential subsets grows exponentially with the acquisition size $b$ and the size of available points to choose from. Instead, we develop a greedy algorithm that selects a batch in linear time, and show that it is at worst a $1 - 1/e$ approximation to the optimal choice for our acquisition function. We provide an open-source implementation[2].

The main contributions of this work are:

1. *BatchBALD*, a data-efficient active learning method that acquires *sets* of high-dimensional image data, leading to improved data efficiency and reduced total run time, section 3.1;

2. a greedy algorithm to select a batch of points efficiently, section 3.2; and

3. an estimator for the acquisition function that scales to larger acquisition sizes and to datasets with many classes, section 3.3.

## 2 Background

### 2.1 Problem Setting

The Bayesian active learning setup consists of an unlabelled dataset $\mathcal{D}_{\textbf{pool}}$, the current training set $\mathcal{D}_{\textbf{train}}$, a Bayesian model $\mathcal{M}$ with model parameters $\boldsymbol{\omega} \sim \mathrm{p}(\boldsymbol{\omega} \mid \mathcal{D}_{\textbf{train}})$, and output predictions $\mathrm{p}(y \mid \boldsymbol{x}, \boldsymbol{\omega}, \mathcal{D}_{\textbf{train}})$ for data point $\boldsymbol{x}$ and prediction $y \in \{1, ..., c\}$ in the classification case. The conditioning of $\boldsymbol{\omega}$ on $\mathcal{D}_{\textbf{train}}$ expresses that the model has been trained with $\mathcal{D}_{\textbf{train}}$. Furthermore, an oracle can provide us with the correct label $\tilde{y}$ for a data point in the unlabelled pool $\boldsymbol{x} \in \mathcal{D}_{\textbf{pool}}$. The goal is to obtain a certain level of prediction accuracy with the least amount of oracle queries. At each acquisition step,

a batch of data points $\left\{ \boldsymbol{x}_1^*, ..., \boldsymbol{x}_b^* \right\}$ is selected using an acquisition function $a$ which scores a candidate batch of unlabelled data points $\{\boldsymbol{x}_1, ..., \boldsymbol{x}_b\} \subseteq \mathcal{D}_{\textbf{pool}}$ using the current model parameters $\mathrm{p}(\boldsymbol{\omega} \mid \mathcal{D}_{\textbf{train}})$:

$$\left\{ \boldsymbol{x}_1^*, ..., \boldsymbol{x}_b^* \right\} = \operatorname*{arg\,max}_{\{\boldsymbol{x}_1, ..., \boldsymbol{x}_b\} \subseteq \mathcal{D}_{\textbf{pool}}} a\left(\{\boldsymbol{x}_1, ..., \boldsymbol{x}_b\}, \mathrm{p}(\boldsymbol{\omega} \mid \mathcal{D}_{\textbf{train}})\right). \tag{1}$$

## 2.2 BALD

BALD (*Bayesian Active Learning by Disagreement*) [19] uses an acquisition function that estimates the mutual information between the model predictions and the model parameters. Intuitively, it captures how strongly the model predictions for a given data point and the model parameters are coupled, implying that finding out about the true label of data points with high mutual information would also inform us about the true model parameters. Originally introduced outside the context of deep learning, the only requirement on the model is that it is Bayesian. BALD is defined as:

$$\mathbb{I}(y ; \boldsymbol{\omega} \mid \boldsymbol{x}, \mathcal{D}_{\textbf{train}}) = \mathbb{H}(y \mid \boldsymbol{x}, \mathcal{D}_{\textbf{train}}) - \mathbb{E}_{\mathrm{p}(\boldsymbol{\omega} \mid \mathcal{D}_{\textbf{train}})} \left[ \mathbb{H}(y \mid \boldsymbol{x}, \boldsymbol{\omega}, \mathcal{D}_{\textbf{train}}) \right]. \tag{2}$$

Looking at the two terms in equation (2), for the mutual information to be high, the left term has to be high and the right term low. The left term is the entropy of the model prediction, which is high when the model's prediction is uncertain. The right term is an expectation of the entropy of the model prediction over the posterior of the model parameters and is low when the model is overall certain for each draw of model parameters from the posterior. Both can only happen when the model has many possible ways to explain the data, which means that the posterior draws are disagreeing among themselves.

BALD was originally intended for acquiring individual data points and immediately retraining the model. This becomes a bottleneck in deep learning, where retraining takes a substantial amount of time. Applications of BALD [12, 20] usually acquire the top $b$. This can be expressed as summing over individual scores:

$$a_{\text{BALD}}\left(\{\boldsymbol{x}_1, ..., \boldsymbol{x}_b\}, \mathrm{p}(\boldsymbol{\omega} \mid \mathcal{D}_{\textbf{train}})\right) = \sum_{i=1}^{b} \mathbb{I}(y_i ; \boldsymbol{\omega} \mid \boldsymbol{x}_i, \mathcal{D}_{\textbf{train}}), \tag{3}$$

and finding the optimal batch for this acquisition function using a greedy algorithm, which reduces to picking the top $b$ highest-scoring data points.

## 2.3 Bayesian Neural Networks (BNN)

In this paper we focus on BNNs as our Bayesian model because they scale well to high dimensional inputs, such as images. Compared to regular neural networks, BNNs maintain a distribution over their weights instead of point estimates. Performing exact inference in BNNs is intractable for any reasonably sized model, so we resort to using a variational approximation. Similar to Gal et al. [13], we use MC dropout [12], which is easy to implement, scales well to large models and datasets, and is straightforward to optimise.

# 3 Methods

## 3.1 BatchBALD

We propose *BatchBALD* as an extension of BALD whereby we jointly score points by estimating the mutual information between a *joint of multiple data points* and the model parameters:[3]

$$a_{\text{BatchBALD}}\left(\{\boldsymbol{x}_1, ..., \boldsymbol{x}_b\}, \mathrm{p}(\boldsymbol{\omega} \mid \mathcal{D}_{\textbf{train}})\right) = \mathbb{I}(y_1, ..., y_b ; \boldsymbol{\omega} \mid \boldsymbol{x}_1, ..., \boldsymbol{x}_b, \mathcal{D}_{\textbf{train}}). \tag{4}$$

This builds on the insight that independent selection of a batch of data points leads to data inefficiency as correlations between data points in an acquisition batch are not taken into account.

To understand how to compute the mutual information between a set of points and the model parameters, we express $\boldsymbol{x}_1, ..., \boldsymbol{x}_b$, and $y_1, ..., y_b$ through joint random variables $\boldsymbol{x}_{1:b}$ and $y_{1:b}$ in a product probability space and use the definition of the mutual information for two random variables:

$$\mathbb{I}(y_{1:b} ; \boldsymbol{\omega} \mid \boldsymbol{x}_{1:b}, \mathcal{D}_{\textbf{train}}) = \mathbb{H}(y_{1:b} \mid \boldsymbol{x}_{1:b}, \mathcal{D}_{\textbf{train}}) - \mathbb{E}_{\mathrm{p}(\boldsymbol{\omega} \mid \mathcal{D}_{\textbf{train}})} \mathbb{H}(y_{1:b} \mid \boldsymbol{x}_{1:b}, \boldsymbol{\omega}, \mathcal{D}_{\textbf{train}}). \tag{5}$$

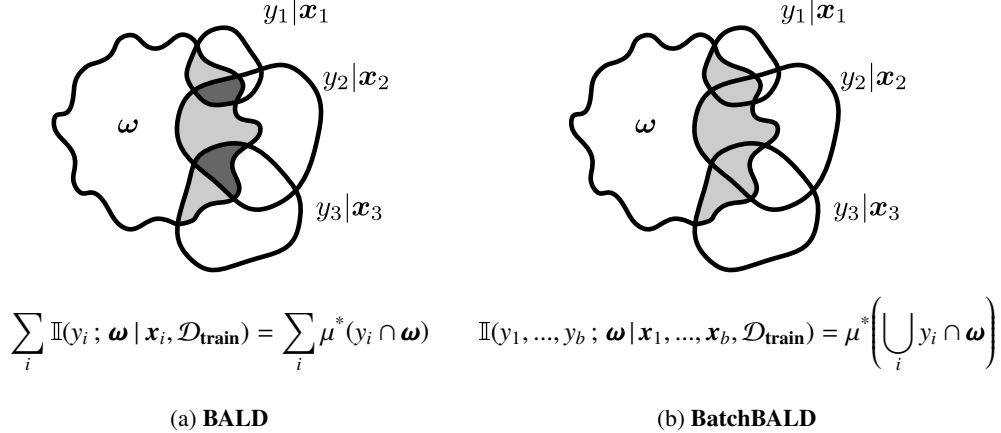

$$\sum_i \mathbb{I}(y_i\,;\,\boldsymbol{\omega}\mid\boldsymbol{x}_i,\mathcal{D}_{\textbf{train}}) = \sum_i \mu^*(y_i\cap\boldsymbol{\omega}) \qquad \mathbb{I}(y_1,...,y_b\,;\,\boldsymbol{\omega}\mid\boldsymbol{x}_1,...,\boldsymbol{x}_b,\mathcal{D}_{\textbf{train}}) = \mu^*\!\left(\bigcup_i y_i\cap\boldsymbol{\omega}\right)$$

(a) **BALD**          (b) **BatchBALD**

Figure 3: *Intuition behind* BALD *and* BatchBALD *using I-diagrams [36]. BALD* overestimates the joint mutual information. *BatchBALD*, however, takes the overlap between variables into account and will strive to acquire a better cover of $\boldsymbol{\omega}$. Areas contributing to the respective score are shown in grey, and areas that are double-counted in dark grey.

---

**Algorithm 1:** Greedy BatchBALD $1 - {}^1\!/_e$-approximate algorithm

    **Input:** acquisition size $b$, unlabelled dataset $\mathcal{D}_{\textbf{pool}}$, model parameters $\mathrm{p}(\boldsymbol{\omega}\mid\mathcal{D}_{\textbf{train}})$
1  $A_0 \leftarrow \emptyset$
2  **for** $n \leftarrow 1$ **to** $b$ **do**
3      **foreach** $\boldsymbol{x} \in \mathcal{D}_{\textit{pool}} \setminus A_{n-1}$ **do** $s_{\boldsymbol{x}} \leftarrow a_{\texttt{BatchBALD}}\left(A_{n-1} \cup \{\boldsymbol{x}\}, \mathrm{p}(\boldsymbol{\omega}\mid\mathcal{D}_{\textbf{train}})\right)$
4      $\boldsymbol{x}_n \leftarrow \underset{\boldsymbol{x}\in\mathcal{D}_{\textbf{pool}}\setminus A_{n-1}}{\arg\max}\; s_{\boldsymbol{x}}$
5      $A_n \leftarrow A_{n-1} \cup \{\boldsymbol{x}_n\}$
6  **end**
    **Output:** acquisition batch $A_n = \{\boldsymbol{x}_1,...,\boldsymbol{x}_b\}$

---

Intuitively, the mutual information between two random variables can be seen as the intersection of their information content. In fact, Yeung [36] shows that a signed measure $\mu^*$ can be defined for discrete random variables $x$, $y$, such that $\mathbb{I}(x\,;\,y) = \mu^*(x\cap y)$, $\mathbb{H}(x,y)=\mu^*(x\cup y)$, $\mathbb{E}_{\mathrm{p}(y)}\,\mathbb{H}(x\mid y) = \mu^*(x\setminus y)$, and so on, where we identify random variables with their counterparts in information space, and conveniently drop conditioning on $\mathcal{D}_{\textbf{train}}$ and $\boldsymbol{x}_i$.

Using this, BALD can be viewed as the sum of individual intersections $\sum_i \mu^*(y_i\cap\boldsymbol{\omega})$, which double counts overlaps between the $y_i$. Naively extending BALD to the mutual information between $y_1,...,y_b$ and $\boldsymbol{\omega}$, which is equivalent to $\mu^*(\bigcap_i y_i\cap\boldsymbol{\omega})$, would lead to selecting *similar* data points instead of diverse ones under maximisation.

BatchBALD, on the other hand, takes overlaps into account by computing $\mu^*(\bigcup_i y_i\cap\boldsymbol{\omega})$ and is more likely to acquire a more diverse cover under maximisation:

$$\mathbb{I}(y_1,...,y_b\,;\,\boldsymbol{\omega}\mid\boldsymbol{x}_1,...,\boldsymbol{x}_b,\mathcal{D}_{\textbf{train}}) = \mathbb{H}(y_{1:b}\mid\boldsymbol{x}_{1:b},\mathcal{D}_{\textbf{train}}) - \mathbb{E}_{\mathrm{p}(\boldsymbol{\omega}\mid\mathcal{D}_{\textbf{train}})}\,\mathbb{H}(y_{1:b}\mid\boldsymbol{x}_{1:b},\boldsymbol{\omega},\mathcal{D}_{\textbf{train}}) \quad (6)$$

$$= \mu^*\!\left(\bigcup_i y_i\right) - \mu^*\!\left(\bigcup_i y_i\setminus\boldsymbol{\omega}\right) = \mu^*\!\left(\bigcup_i y_i\cap\boldsymbol{\omega}\right) \quad (7)$$

This is depicted in figure 3 and also motivates that $a_{\texttt{BatchBALD}} \le a_{\texttt{BALD}}$, which we prove in appendix B.1. For acquisition size 1, BatchBALD and BALD are equivalent.

### 3.2 Greedy approximation algorithm for BatchBALD

To avoid the combinatorial explosion that arises from jointly scoring subsets of points, we introduce a greedy approximation for computing BatchBALD, depicted in algorithm 1. In appendix A, we prove that $a_{\texttt{BatchBALD}}$ is submodular, which means the greedy algorithm is $1 - {}^1\!/_e$-approximate [8, 24, 25].

In appendix B.2, we show that, under idealised conditions, when using BatchBALD and a fixed final $|\mathcal{D}_{\textbf{train}}|$, the active learning loop itself can be seen as a greedy $1 - 1/e$-approximation algorithm, and that an active learning loop with BatchBALD and acquisition size larger than 1 is bounded by an an active learning loop with individual acquisitions, that is BALD/BatchBALD with acquisition size 1, which is the ideal case.

### 3.3 Computing $a_{\texttt{BatchBALD}}$

For brevity, we leave out conditioning on $x_1, ..., x_n$, and $\mathcal{D}_{\textbf{train}}$, and $\mathrm{p}(\boldsymbol{\omega})$ denotes $\mathrm{p}(\boldsymbol{\omega} \mid \mathcal{D}_{\textbf{train}})$ in this section. $a_{\texttt{BatchBALD}}$ is then written as:

$$a_{\texttt{BatchBALD}}\left(\{x_1, ..., x_n\}, \mathrm{p}(\boldsymbol{\omega})\right) = \mathbb{H}(y_1, ..., y_n) - \mathbb{E}_{\mathrm{p}(\boldsymbol{\omega})}\left[\mathbb{H}(y_1, ..., y_n \mid \boldsymbol{\omega})\right]. \tag{8}$$

Because the $y_i$ are independent when conditioned on $\boldsymbol{\omega}$, computing the right term of equation (8) is simplified as the conditional joint entropy decomposes into a sum. We can approximate the expectation using a Monte-Carlo estimator with $k$ samples from our model parameter distribution $\hat{\boldsymbol{\omega}}_j \sim \mathrm{p}(\boldsymbol{\omega})$:

$$\mathbb{E}_{\mathrm{p}(\boldsymbol{\omega})}\left[\mathbb{H}(y_1, ..., y_n \mid \boldsymbol{\omega})\right] = \sum_{i=1}^{n} \mathbb{E}_{\mathrm{p}(\boldsymbol{\omega})}\left[\mathbb{H}(y_i \mid \boldsymbol{\omega})\right] \approx \frac{1}{k}\sum_{i=1}^{n}\sum_{j=1}^{k}\mathbb{H}(y_i \mid \hat{\boldsymbol{\omega}}_j). \tag{9}$$

Computing the left term of equation (8) is difficult because the unconditioned joint probability does not factorise. Applying the equality $\mathrm{p}(y) = \mathbb{E}_{\mathrm{p}(\boldsymbol{\omega})}\left[\mathrm{p}(y \mid \boldsymbol{\omega})\right]$, and, using sampled $\hat{\boldsymbol{\omega}}_j$, we compute the entropy by summing over all possible configurations $\hat{y}_{1:n}$ of $y_{1:n}$:

$$\mathbb{H}(y_1, ..., y_n) = \mathbb{E}_{\mathrm{p}(y_1, ..., y_n)}\left[-\log \mathrm{p}(y_1, ..., y_n)\right] \tag{10}$$

$$= \mathbb{E}_{\mathrm{p}(\boldsymbol{\omega})}\,\mathbb{E}_{\mathrm{p}(y_1, ..., y_n \mid \boldsymbol{\omega})}\left[-\log \mathbb{E}_{\mathrm{p}(\boldsymbol{\omega})}\left[\mathrm{p}(y_1, ..., y_n \mid \boldsymbol{\omega})\right]\right] \tag{11}$$

$$\approx -\sum_{\hat{y}_{1:n}}\left(\frac{1}{k}\sum_{j=1}^{k}\mathrm{p}(\hat{y}_{1:n} \mid \hat{\boldsymbol{\omega}}_j)\right)\log\left(\frac{1}{k}\sum_{j=1}^{k}\mathrm{p}(\hat{y}_{1:n} \mid \hat{\boldsymbol{\omega}}_j)\right). \tag{12}$$

### 3.4 Efficient implementation

In each iteration of the algorithm, $x_1, ..., x_{n-1}$ stay fixed while $x_n$ varies over $\mathcal{D}_{\textbf{pool}} \setminus A_{n-1}$. We can reduce the required computations by factorizing $\mathrm{p}(y_{1:n} \mid \boldsymbol{\omega})$ into $\mathrm{p}(y_{1:n-1} \mid \boldsymbol{\omega})\,\mathrm{p}(y_n \mid \boldsymbol{\omega})$. We store $\mathrm{p}(\hat{y}_{1:n-1} \mid \hat{\boldsymbol{\omega}}_j)$ in a matrix $\hat{P}_{1:n-1}$ of shape $c^{n-1} \times k$ and $\mathrm{p}(y_n \mid \hat{\boldsymbol{\omega}}_j)$ in a matrix $\hat{P}_n$ of shape $c \times k$. The sum $\sum_{j=1}^{k}\mathrm{p}(\hat{y}_{1:n} \mid \hat{\boldsymbol{\omega}}_j)$ in (12) can be then be turned into a matrix product:

$$\frac{1}{k}\sum_{j=1}^{k}\mathrm{p}(\hat{y}_{1:n} \mid \hat{\boldsymbol{\omega}}_j) = \frac{1}{k}\sum_{j=1}^{k}\mathrm{p}(\hat{y}_{1:n-1} \mid \hat{\boldsymbol{\omega}}_j)\,\mathrm{p}(\hat{y}_n \mid \hat{\boldsymbol{\omega}}_j) = \left(\frac{1}{k}\hat{P}_{1:n-1}\hat{P}_n^T\right)_{\hat{y}_{1:n-1}, \hat{y}_n}. \tag{13}$$

This can be further sped up by using batch matrix multiplication to compute the joint entropy for different $x_n$. $\hat{P}_{1:n-1}$ only has to be computed once, and we can recursively compute $\hat{P}_{1:n}$ using $\hat{P}_{1:n-1}$ and $\hat{P}_n$, which allows us to sample $\mathrm{p}(y \mid \hat{\boldsymbol{\omega}}_j)$ for each $x \in \mathcal{D}_{\textbf{pool}}$ only once at the beginning of the algorithm.

For larger acquisition sizes, we use $m$ MC samples of $y_{1:n-1}$ as enumerating all possible configurations becomes infeasible. See appendix C for details.

Monte-Carlo sampling bounds the time complexity of the full BatchBALD algorithm to $O(bc \cdot \min\{c^b, m\} \cdot |\mathcal{D}_{\textbf{pool}}| \cdot k)$ compared to $O(c^b \cdot |\mathcal{D}_{\textbf{pool}}|^b \cdot k)$ for naively finding the exact optimal batch and $O((b + k) \cdot |\mathcal{D}_{\textbf{pool}}|)$ for BALD[4].

## 4 Experiments

In our experiments, we start by showing how a naive application of the BALD algorithm to an image dataset can lead to poor results in a dataset with many (near) duplicate data points, and show that BatchBALD solves this problem in a grounded way while obtaining favourable results (figure 2).

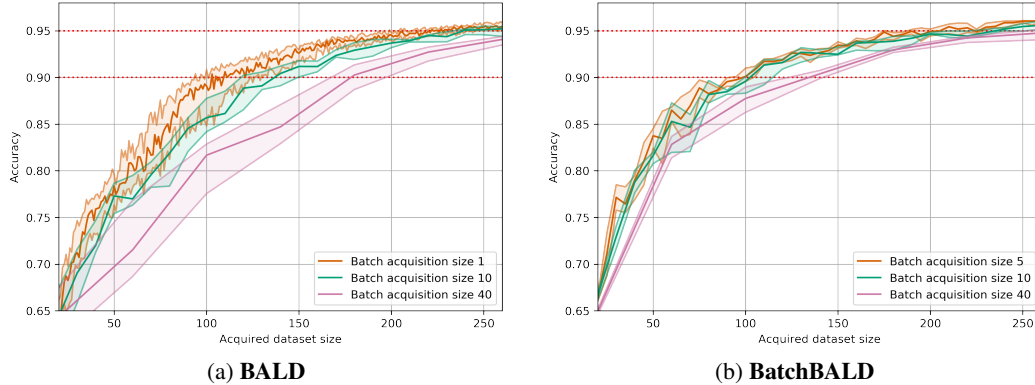

(a) **BALD**                (b) **BatchBALD**

Figure 4: *Performance on* MNIST *for increasing acquisition sizes.* BALD's performance drops drastically as the acquisition size increases. BatchBALD maintains strong performance even with increasing acquisition size.

We then illustrate BatchBALD's effectiveness on standard AL datasets: MNIST and EMNIST. EMNIST [6] is an extension of MNIST that also includes letters, for a total of 47 classes, and has a twice as large training set. See appendix F for examples of the dataset. We show that BatchBALD provides a substantial performance improvement in these scenarios, too, and has more diverse acquisitions. Finally, we look at BatchBALD in the setting of transfer learning, where we finetune a large pretrained model on a more difficult dataset called CINIC-10 [9], which is a combination of CIFAR-10 and downscaled ImageNet.

In our experiments, we repeatedly go through active learning loops. One active learning loop consists of training the model on the available labelled data and subsequently acquiring new data points using a chosen acquisition function. As the labelled dataset is small in the beginning, it is important to avoid overfitting. We do this by using early stopping after 3 epochs of declining accuracy on the validation set. We pick the model with the highest validation accuracy. Throughout our experiments, we use the Adam [22] optimiser with learning rate 0.001 and betas 0.9/0.999. All our results report the median of 6 trials, with lower and upper quartiles. We use these quartiles to draw the filled error bars on our figures.

We reinitialize the model after each acquisition, similar to Gal et al. [13]. We found this helps the model improve even when very small batches are acquired. It also decorrelates subsequent acquisitions as final model performance is dependent on a particular initialization [10].

When computing $p(y|x, \boldsymbol{\omega}, \mathcal{D}_{\mathbf{train}})$, it is important to keep the dropout masks in MC dropout consistent while sampling from the model. This is necessary to capture dependencies between the inputs for BatchBALD, and it makes the scores for different points more comparable by removing this source of noise. We do not keep the masks fixed when computing BALD scores because its performance usually benefits from the added noise. We also do not need to keep these masks fixed for training and evaluating the model.

In all our experiments, we either compute joint entropies exactly by enumerating all configurations, or we estimate them using 10,000 MC samples, picking whichever method is faster. In practice, we compute joint entropies exactly for roughly the first 4 data points in an acquisition batch and use MC sampling thereafter.

## 4.1 Repeated MNIST

As demonstrated in the introduction, naively applying BALD to a dataset that contains many (near) replicated data points leads to poor performance. We show how this manifests in practice by taking the MNIST dataset and replicating each data point in the training set two times (obtaining a training set that is three times larger than the original). After normalising the dataset, we add isotropic Gaussian noise with a standard deviation of 0.1 to simulate slight differences between the duplicated data points in the training set. All results are obtained using an acquisition size of 10 and 10 MC

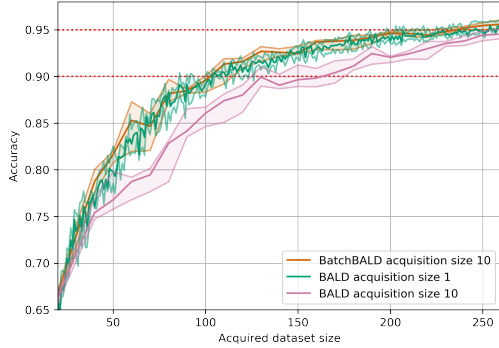
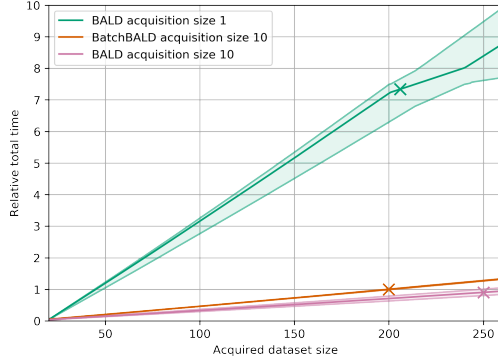

Figure 5: *Performance on* MNIST. BatchBALD outperforms BALD with acquisition size 10 and performs close to the optimum of acquisition size 1.

Figure 6: *Relative total time on* MNIST. Normalized to training BatchBALD with acquisition size 10 to 95% accuracy. The stars mark when 95% accuracy is reached for each method.

Table 1: *Number of required data points on* MNIST *until 90% and 95% accuracy are reached.* 25%-, 50%- and 75%-quartiles for the number of required data points when available.

|  | 90% accuracy | 95% accuracy |
|---|---|---|
| BatchBALD | 70 / 90 / 110 | 190 / 200 / 230 |
| BALD [6] | 120 / 120 / 170 | 250 / 250 / >300 |
| BALD [13] | 145 | 335 |

dropout samples. The initial dataset was constructed by taking a balanced set of 20 data points[5], two of each class (similar to [13]).

Our model consists of two blocks of [convolution, dropout, max-pooling, relu], with 32 and 64 5x5 convolution filters. These blocks are followed by a two-layer MLP that includes dropout between the layers and has 128 and 10 hidden units. The dropout probability is 0.5 in all three locations. This architecture achieves 99% accuracy with 10 MC dropout samples during test time on the full MNIST dataset.

The results can be seen in figure 2. In this illustrative scenario, BALD performs poorly, and even randomly acquiring points performs better. However, BatchBALD is able to cope with the replication perfectly. In appendix D, we look at varying the repetition number and show that as we increase the number of repetitions BALD gradually performs worse. In appendix E, we also compare with Variation Ratios [11], and Mean STD [21] which perform on par with random acquisition.

## 4.2 MNIST

For the second experiment, we follow the setup of Gal et al. [13] and perform AL on the MNIST dataset using 100 MC dropout samples. We use the same model architecture and initial dataset as described in section 4.1. Due to differences in model architecture, hyper parameters and model retraining, we significantly outperform the original results in Gal et al. [13] as shown in table 1.

We first look at BALD for increasing acquisition size in figure 4a. As we increase the acquisition size from the ideal of acquiring points individually and fully retraining after each points (acquisition size 1) to 40, there is a substantial performance drop.

BatchBALD, in figure 4b, is able to maintain performance when doubling the acquisition size from 5 to 10. Performance drops only slightly at 40, possibly due to estimator noise.

The results for acquisition size 10 for both BALD and BatchBALD are compared in figure 5. BatchBALD outperforms BALD. Indeed, BatchBALD with acquisition size 10 performs close to the ideal with acquisition size 1. The total run time of training these three models until 95% accuracy is

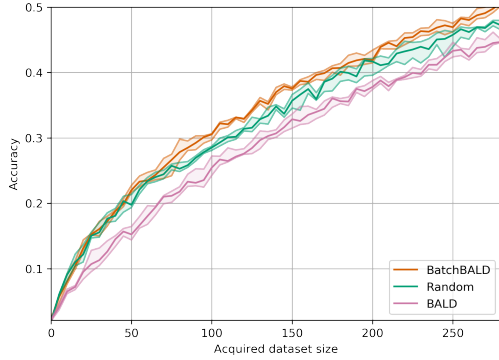

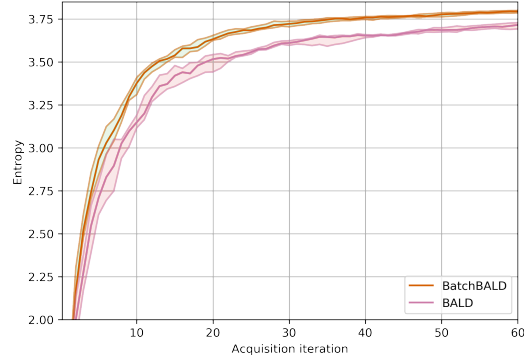

Figure 7: *Performance on* EMNIST. BatchBALD consistently outperforms both random acquisition and BALD while BALD is unable to beat random acquisition.

Figure 8: *Entropy of acquired class labels over acquisition steps on* EMNIST. BatchBALD steadily acquires a more diverse set of data points.

visualized in figure 6, where we see that BatchBALD with acquisition size 10 is much faster than BALD with acquisition size 1, and only marginally slower than BALD with acquisition size 10.

### 4.3 EMNIST

In this experiment, we show that BatchBALD also provides a significant improvement when we consider the more difficult EMNIST dataset [6] in the *Balanced* setup, which consists of 47 classes, comprising letters and digits. The training set consists of 112,800 28x28 images balanced by class, of which the last 18,800 images constitute the validation set. We do not use an initial dataset and instead perform the initial acquisition step with the randomly initialized model. We use 10 MC dropout samples.

We use a similar model architecture as before, but with added capacity. Three blocks of [convolution, dropout, max-pooling, relu], with 32, 64 and 128 3x3 convolution filters, and 2x2 max pooling. These blocks are followed by a two-layer MLP with 512 and 47 hidden units, with again a dropout layer in between. We use dropout probability 0.5 throughout the model.

The results for acquisition size 5 can be seen in figure 7. BatchBALD outperforms both random acquisition and BALD while BALD is unable to beat random acquisition. Figure 8 gives some insight into why BatchBALD performs better than BALD. The entropy of the categorical distribution of acquired class labels is consistently higher, meaning that BatchBALD acquires a more diverse set of data points. In figure 15, the classes on the x-axis are sorted by number of data points that were acquired of that class. We see that BALD undersamples classes while BatchBALD is more consistent.

### 4.4 CINIC-10

CINIC-10 is an interesting dataset because it is large (270k data points) and its data comes from two different sources: CIFAR-10 and ImageNet. To get strong performance on the test set it is important to obtain data from both sets. Instead of training a very deep model from scratch on a small dataset, we opt to run this experiment in a transfer learning setting, where we use a pretrained model and acquire data only to finetune the original model. This is common practice and suitable in cases where data is abound for an auxiliary domain, but is expensive to label for the domain of interest.

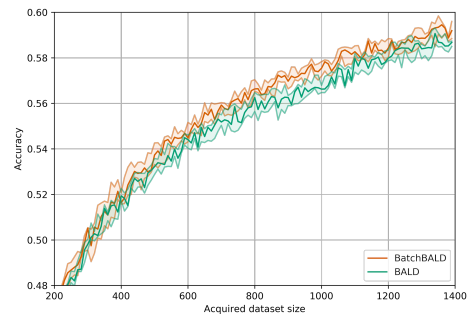

Figure 9: *Performance on* CINIC-10. BatchBALD outperforms BALD from 500 acquired samples onwards.

For the CINIC-10 experiment, we use 160k training samples for the unlabelled pool, 20k validation samples, and the other 90k as test samples. We use an

ImageNet pretrained VGG-16, provided by PyTorch [26], with a dropout layer before a 512 hidden unit (instead of 4096) fully connected layer. We use 50 MC dropout samples, acquisition size 10 and repeat the experiment for 6 trials. The results are in figure 9, with the 59% mark reached at 1170 for BatchBALD and 1330 for BALD (median).

## 5   Related work

AL is closely related to Bayesian Optimisation (BO), which is concerned with finding the global optimum of a function [33], with the fewest number of function evaluations. This is generally done using a Gaussian Process. A common problem in BO is the lack of parallelism, with usually a single worker being responsible for function evaluations. In real-world settings, there are usually many such workers available and making optimal use of them is an open problem [14, 2] with some work exploring mutual information for optimising a multi-objective problem [17].

Maintaining diversity when acquiring a batch of data has also been attempted using constrained optimisation [15] and in Gaussian Mixture Models [3]. In AL of molecular data, the lack of diversity in batches of data points acquired using the BALD objective has been noted by Janz et al. [20], who propose to resolve it by limiting the number of MC dropout samples and relying on noisy estimates.

A related approach to AL is semi-supervised learning (also sometimes referred to as weakly-supervised), in which the labelled data is commonly assumed to be fixed and the unlabelled data is used for unsupervised learning [23, 27]. Wang et al. [35], Sener and Savarese [29], Samarth Sinha [28] explore combining it with AL.

## 6   Scope and limitations

**Unbalanced datasets** BALD and BatchBALD do not work well when the test set is unbalanced as they aim to learn well about all classes and do not follow the density of the dataset. However, if the test set is balanced, but the training set is not, we expect BatchBALD to perform well.

**Unlabelled data** BatchBALD does not take into account any information from the unlabelled dataset. However, BatchBALD uses the underlying Bayesian model for estimating uncertainty for unlabelled data points, and semi-supervised learning could improve these estimates by providing more information about the underlying structure of the feature space. We leave a semi-supervised extension of BatchBALD to future work.

**Noisy estimator** A significant amount of noise is introduced by MC-dropout's variational approximation to training BNNs. Sampling of the joint entropies introduces additional noise. The quality of larger acquisition batches would be improved by reducing this noise.

## 7   Conclusion

We have introduced a new batch acquisition function, BatchBALD, for Deep Bayesian Active Learning, and a greedy algorithm that selects good candidate batches compared to the intractable optimal solution. Acquisitions show increased diversity of data points and improved performance over BALD and other methods.

While our method comes with additional computational cost during acquisition, BatchBALD is able to significantly reduce the number of data points that need to be labelled and the number of times the model has to be retrained, potentially saving considerable costs and filling an important gap in practical Deep Bayesian Active Learning.

**Acknowledgements**

The authors want to thank Binxin (Robin) Ru for helpful references to submodularity and the appropriate proofs. We would also like to thank the rest of OATML for their feedback at several stages of the project. AK is supported by the UK EPSRC CDT in Autonomous Intelligent Machines and Systems (grant reference EP/L015897/1). JvA is grateful for funding by the EPSRC (grant reference EP/N509711/1) and Google-DeepMind. Funding for computational resources was provided by the Allan Turing Institute and Google.

**Author contributions**

AK derived the original estimator, proved submodularity and bounds, implemented BatchBALD efficiently, and ran the experiments. JvA developed the narrative and experimental design, advised on debugging, structured the paper into its current form, and pushed it forward at difficult times. JvA and AK wrote the paper jointly.

## Footnotes

[2]`https://github.com/BlackHC/BatchBALD`

[3]We use the notation $\mathbb{I}(x, y ; z \mid c)$ to denote the mutual information between the *joint of the random variables* $x, y$ and the random variable $z$ conditioned on $c$.

[4]$b$ is the acquisition size, $c$ is the number of classes, $k$ is the number of MC dropout samples, and $m$ is the number of sampled configurations of $y_{1:n-1}$.

[5]These initial data points were chosen by running BALD 6 times with the initial dataset picked randomly and choosing the set of the median model. They were subsequently held fixed.

[6]reimplementation using reported experimental setup

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
