[Supplementary Material]

# A  Proof of submodularity

Nemhauser et al. [25] show that if a function is submodular, then a greedy algorithm like algorithm 1 is $1 - 1/e$-approximate. Here, we show that $a_{\texttt{BatchBALD}}$ is submodular.

We will show that $a_{\texttt{BatchBALD}}$ satisfies the following equivalent definition of submodularity:

**Definition A.1.** *A function $f$ defined on subsets of $\Omega$ is called* submodular *if for every set $A \subset \Omega$ and two non-identical points $y_1, y_2 \in \Omega \setminus A$:*

$$f(A \cup \{y_1\}) + f(A \cup \{y_2\}) \geq f(A \cup \{y_1, y_2\}) + f(A) \tag{14}$$

Submodularity expresses that there are "diminishing returns" for adding additional points to $f$.

**Lemma A.2.** $a_{\texttt{BatchBALD}}(A, p(\boldsymbol{\omega})) := \mathbb{I}(A \,;\, \boldsymbol{\omega})$ *is submodular for* $A \subset \mathcal{D}_{pool}$.

*Proof.* Let $y_1, y_2 \in \mathcal{D}_{\textbf{pool}}, y_1 \neq y_2$. We start by substituting the definition of $a_{\texttt{BatchBALD}}$ into (14) and subtracting $\mathbb{I}(A \,;\, \boldsymbol{\omega})$ twice on both sides, using that $\mathbb{I}(A \cup B \,;\, \boldsymbol{\omega}) - \mathbb{I}(B \,;\, \boldsymbol{\omega}) = \mathbb{I}(A \,;\, \boldsymbol{\omega} \,|\, B)$:

$$\mathbb{I}(A \cup \{y\} \,;\, \boldsymbol{\omega}) + \mathbb{I}(A \cup \{x\} \,;\, \boldsymbol{\omega}) \geq \mathbb{I}(A \cup \{x, y\} \,;\, \boldsymbol{\omega}) + \mathbb{I}(A \,;\, \boldsymbol{\omega}) \tag{15}$$

$$\Leftrightarrow \mathbb{I}(y \,;\, \boldsymbol{\omega} \,|\, A) + \mathbb{I}(x \,;\, \boldsymbol{\omega} \,|\, A) \geq \mathbb{I}(x, y \,;\, \boldsymbol{\omega} \,|\, A). \tag{16}$$

We rewrite the left-hand side using the definition of the mutual information $\mathbb{I}(A \,;\, B) = \mathbb{H}(A) - \mathbb{H}(A \,|\, B)$ and reorder:

$$\mathbb{I}(y \,;\, \boldsymbol{\omega} \,|\, A) + \mathbb{I}(x \,;\, \boldsymbol{\omega} \,|\, A) \tag{17}$$

$$= \underbrace{\mathbb{H}(y_1 \,|\, A) + \mathbb{H}(y_1 \,|\, A)}_{\geq \mathbb{H}(y_1, y_2 | A)} - \underbrace{(\mathbb{H}(y_1 \,|\, A, \boldsymbol{\omega}) + \mathbb{H}(y_2 \,|\, A, \boldsymbol{\omega}))}_{= \mathbb{H}(y_1, y_2 | A, \boldsymbol{\omega})} \tag{18}$$

$$\geq \mathbb{H}(y_1, y_2 \,|\, A) - \mathbb{H}(y_1, y_2 \,|\, A, \boldsymbol{\omega}) \tag{19}$$

$$= \mathbb{I}(x, y \,;\, \boldsymbol{\omega} \,|\, A), \tag{20}$$

where we have used that entropies are subadditive in general and additive given $y_1 \perp\!\!\!\perp y_2 \,|\, \boldsymbol{\omega}$. □

Following Nemhauser et al. [25], we can conclude that algorithm 1 is $1 - 1/e$-approximate.

# B  Connection between BatchBALD and BALD

In the following section, we show that BALD approximates BatchBALD and that BatchBALD approximates BALD with acquisition size 1. The BALD score is an upper bound of the BatchBALD score for any candidate batch. At the same time, BatchBALD can be seen as performing BALD with acquisition size 1 during each step of its greedy algorithm in an idealised setting.

## B.1  BALD as an approximation of BatchBALD

Using the subadditivity of information entropy and the independence of the $y_i$ given $\boldsymbol{\omega}$, we show that BALD is an approximation of BatchBALD and is always an upper bound on the respective BatchBALD score:

$$a_{\texttt{BatchBALD}} (\{\boldsymbol{x}_1, ..., \boldsymbol{x}_b\}, p(\boldsymbol{\omega} \,|\, \mathcal{D}_{\textbf{train}})) \tag{21}$$

$$= \mathbb{H}(y_1, ..., y_b \,|\, \boldsymbol{x}_1, ..., \boldsymbol{x}_b, \mathcal{D}_{\textbf{train}}) - \mathbb{E}_{p(\boldsymbol{\omega}|\mathcal{D}_{\textbf{train}})} \left[ \mathbb{H}(y_1, ..., y_b \,|\, \boldsymbol{x}_1, ..., \boldsymbol{x}_b, \boldsymbol{\omega}, \mathcal{D}_{\textbf{train}}) \right] \tag{22}$$

$$\leq \sum_{i=1}^{b} \mathbb{H}(y_i \,|\, \boldsymbol{x}_i, \mathcal{D}_{\textbf{train}}) - \sum_{i=1}^{b} \mathbb{E}_{p(\boldsymbol{\omega}|\mathcal{D}_{\textbf{train}})} \left[ \mathbb{H}(y_i \,|\, \boldsymbol{x}_i, \boldsymbol{\omega}, \mathcal{D}_{\textbf{train}}) \right] \tag{23}$$

$$= \sum_{i=1}^{b} \mathbb{I}(y_i \,;\, \boldsymbol{\omega} \,|\, \boldsymbol{x}_i, \mathcal{D}_{\textbf{train}}) = a_{\texttt{BALD}} (\{\boldsymbol{x}_1, ..., \boldsymbol{x}_b\}, p(\boldsymbol{\omega} \,|\, \mathcal{D}_{\textbf{train}})) \tag{24}$$

## B.2 BatchBALD as an approximation of BALD with acquisition size 1

To see why BALD with acquisition size 1 can be seen as an upper bound for BatchBALD performance in an idealised setting, we reformulate line 3 in algorithm 1 on page 4.

Instead of the original term $a_{\texttt{BatchBALD}}\left(A_{n-1} \cup \{x\}, \mathrm{p}(\boldsymbol{\omega} \mid \mathcal{D}_{\textbf{train}})\right)$, we can equivalently maximise

$$a_{\texttt{BatchBALD}}\left(A_{n-1} \cup \{x\}, \mathrm{p}(\boldsymbol{\omega} \mid \mathcal{D}_{\textbf{train}})\right) - a_{\texttt{BatchBALD}}\left(A_{n-1}, \mathrm{p}(\boldsymbol{\omega} \mid \mathcal{D}_{\textbf{train}})\right) \qquad (25)$$

as the right term is constant for all $x \in \mathcal{D}_{\textbf{pool}} \setminus A_{n-1}$ within the inner loop, which, in turn, is equivalent to

$$= \mathbb{I}(y_1, ..., y_{n-1}, y\,;\, \boldsymbol{\omega} \mid x_1, ..., x_{n-1}, x, \mathcal{D}_{\textbf{train}}) - \mathbb{I}(y_1, ..., y_{n-1}\,;\, \boldsymbol{\omega} \mid x_{1:n-1}\mathcal{D}_{\textbf{train}}) \qquad (26)$$

$$= \mathbb{I}(y\,;\, \boldsymbol{\omega} \mid x, y_1, ..., y_{n-1}, x_{1:n-1}, \mathcal{D}_{\textbf{train}}) \qquad (27)$$

once we expand $A_{n-1} = \{x_1, ..., x_{n-1}\}$. This means that, at each step of the inner loop, our greedy algorithm is maximising the mutual information of the individual available data points with the model parameters conditioned on all the additional data points that have already been picked for acquisition and the existing training set. Finally, assuming training our model captures all available information,

$$\geq \mathbb{I}(y\,;\, \boldsymbol{\omega} \mid x, \mathcal{D}_{\textbf{train}} \cup \{(x_1, \tilde{y}_1), ...., (x_{n-1}, \tilde{y}_{n-1})\}) \qquad (28)$$

$$= a_{\texttt{BALD}}\left(\{x\}, \mathrm{p}(\boldsymbol{\omega} \mid \mathcal{D}_{\textbf{train}} \cup \{(x_1, \tilde{y}_1), ...., (x_{n-1}, \tilde{y}_{n-1})\})\right), \qquad (29)$$

where $\tilde{y}_1, ..., \tilde{y}_{n-1}$ are the actual labels of $x_1, ..., x_n$. The mutual information decreases as $\boldsymbol{\omega}$ becomes more concentrated as we expand its training set, and thus the overlap of $y$ and $\boldsymbol{\omega}$ will become smaller (in an information-measure-theoretical sense).

This shows that every step $n$ of the inner loop in our algorithm is at most as good as retraining our model on the new training set $\mathcal{D}_{\textbf{train}} \cup \{(x_1, \tilde{y}_1), ...., (x_{n-1}, \tilde{y}_{n-1})\}$ and picking $x_n$ using $a_{\texttt{BALD}}$ with acquisition size 1.

**Relevance for the active training loop.** We see that the active training loop as a whole is computing a greedy $1 - 1/e$-approximation of the mutual information of all acquired data points over all acquisitions with the model parameters.

## C  Sampling of configurations

We are using the same notation as in section 3.3. We factor $\mathrm{p}(y_{1:n} \mid \boldsymbol{\omega})$ to avoid recomputations and rewrite $\mathbb{H}(y_{1:n})$ as:

$$\mathbb{H}(y_{1:n}) = \mathbb{E}_{\mathrm{p}(\boldsymbol{\omega})}\, \mathbb{E}_{\mathrm{p}(y_{1:n}|\boldsymbol{\omega})}\left[-\log \mathrm{p}(y_{1:n})\right] \qquad (30)$$

$$= \mathbb{E}_{\mathrm{p}(\boldsymbol{\omega})}\, \mathbb{E}_{\mathrm{p}(y_{1:n-1}|\boldsymbol{\omega})\,\mathrm{p}(y_n|\boldsymbol{\omega})}\left[-\log \mathrm{p}(y_{1:n})\right] \qquad (31)$$

$$= \mathbb{E}_{\mathrm{p}(\boldsymbol{\omega})}\, \mathbb{E}_{\mathrm{p}(y_{1:n-1}|\boldsymbol{\omega})}\, \mathbb{E}_{\mathrm{p}(y_n|\boldsymbol{\omega})}\left[-\log \mathrm{p}(y_{1:n})\right] \qquad (32)$$

To be flexible in the way we sample $y_{1:n-1}$, we perform importance sampling of $\mathrm{p}(y_{1:n-1} \mid \boldsymbol{\omega})$ using $\mathrm{p}(y_{1:n-1})$, and, assuming we also have $m$ samples $\hat{y}_{1:n-1}$ from $\mathrm{p}(y_{1:n-1})$, we can approximate:

$$\mathbb{H}(y_{1:n}) = \mathbb{E}_{\mathrm{p}(\boldsymbol{\omega})}\, \mathbb{E}_{\mathrm{p}(y_{1:n-1})}\left[\frac{\mathrm{p}(y_{1:n-1} \mid \boldsymbol{\omega})}{\mathrm{p}(y_{1:n-1})}\, \mathbb{E}_{\mathrm{p}(y_n|\boldsymbol{\omega})}\left[-\log \mathrm{p}(y_{1:n})\right]\right] \qquad (33)$$

$$= \mathbb{E}_{\mathrm{p}(y_{1:n-1})}\, \mathbb{E}_{\mathrm{p}(\boldsymbol{\omega})}\, \mathbb{E}_{\mathrm{p}(y_n|\boldsymbol{\omega})}\left[-\frac{\mathrm{p}(y_{1:n-1} \mid \boldsymbol{\omega})}{\mathrm{p}(y_{1:n-1})}\log \mathbb{E}_{\mathrm{p}(\boldsymbol{\omega})}\left[\mathrm{p}(y_{1:n-1} \mid \boldsymbol{\omega})\,\mathrm{p}(y_{1:n} \mid \boldsymbol{\omega})\right]\right] \qquad (34)$$

$$\approx -\frac{1}{m}\sum_{\hat{y}_{1:n-1}}^{m}\sum_{\hat{y}_n}\frac{\frac{1}{k}\sum_{\hat{\boldsymbol{\omega}}_j}\mathrm{p}(\hat{y}_{1:n-1} \mid \hat{\boldsymbol{\omega}}_j)\,\mathrm{p}(\hat{y}_n \mid \hat{\boldsymbol{\omega}}_j)}{\mathrm{p}(\hat{y}_{1:n-1})}\log\left(\frac{1}{k}\sum_{\hat{\boldsymbol{\omega}}_j}\mathrm{p}(\hat{y}_{1:n-1} \mid \hat{\boldsymbol{\omega}}_j)\,\mathrm{p}(\hat{y}_n \mid \hat{\boldsymbol{\omega}}_j)\right) \qquad (35)$$

$$= -\frac{1}{m}\sum_{\hat{y}_{1:n-1}}^{m}\sum_{\hat{y}_n}\frac{\left(\hat{P}_{1:n-1}\hat{P}_n^T\right)_{\hat{y}_{1:n-1},\hat{y}_n}}{\left(\hat{P}_{1:n-1}\mathbb{1}_{k,1}\right)_{\hat{y}_{1:n-1}}}\log\left(\frac{1}{k}\left(\hat{P}_{1:n-1}\hat{P}_n^T\right)_{\hat{y}_{1:n-1},\hat{y}_n}\right), \qquad (36)$$

where we store $\mathrm{p}(\hat{y}_{1:n-1} \mid \hat{\boldsymbol{\omega}}_j)$ in a matrix $\hat{P}_{1:n-1}$ of shape $m \times k$ and $\mathrm{p}(\hat{y}_n \mid \hat{\boldsymbol{\omega}}_j)$ in a matrix $\hat{P}_n$ of shape $c \times k$ and $\mathbb{1}_{k,1}$ is a $k \times 1$ matrix of 1s. Equation (36) allows us to cache $\hat{P}_{1:n-1}$ inside the inner loop of algorithm 1 and use batch matrix multiplication for efficient computation.

# D    Ablation study on Repeated MNIST

To better understand the effect of redundant data points on BALD and BatchBALD, we run the RMNIST experiment with an increasing number of repetitions. The results can be seen in figure 10. We use the same setup as in section 4.1. BatchBALD performs the same on all repetition numbers (100 data points till 90%). BALD achieves 90% accuracy at 120 data points (0 repetitions), 160 data points (1 repetition), 280 data points (2 repetitions), 300 data points (4 repetitions). This shows that BALD and BatchBALD behave as expected.

Figure 10: *Performance of* BALD *on* Repeated MNIST *for increasing amount of repetitions.* We see that BALD performs worse as the number of repetitions is increased, while BatchBALD outperforms BALD with zero repetitions.

# E    Additional results for Repeated MNIST

We show that BatchBALD also outperforms Var Ratios [11] and Mean STD [21].

Figure 11: *Performance on* Repeated MNIST. BALD, BatchBALD, Var Ratios, Mean STD and random acquisition with acquisition size 10 and 10 MC dropout samples.

# F  Example visualisation of EMNIST

Figure 12: *Examples of all 47 classes of EMNIST*

# G  Entropy and class acquisitions including random acquisition

Figure 13: *Performance on* EMNIST. Batch-BALD consistently outperforms both random acquisition and BALD while BALD is unable to beat random acquisition.

Figure 14: *Entropy of acquired class labels over acquisition steps on* EMNIST. BatchBALD steadily acquires a more diverse set of data points than BALD.

Figure 15: *Histogram of acquired class labels on* EMNIST. BatchBALD left and BALD right. Classes are sorted by number of acquisitions. Several EMNIST classes are underrepresented in BALD and random acquisition while BatchBALD acquires classes more uniformly. The histograms were created from all acquired points at the end of an active learning loop