[Reviews · NeurIPS 2019]

Reviewer 1



post-rebuttal: Thank you for the clarification. My score remains the same. ----------------------------- It is an interesting work that addresses the degeneracy of batch acquisition when using BALD as a score function. The methods proposed in the paper elegantly deals with the problem of redundant acquisition when using BALD in a greedy manner. I have a few questions and hope the authors can address them: (1) Does this problem of redundant acquisition only happen when one uses BALD as the score? Intuitively I would think no, as if one uses any score function greedily, regardless of the contribution of the other samples selected in the same batch, one can still end up with a biased batch that can potentially harm training. If this is the case, then why are var-ratios and mean-std outperforming random? I guess it is a matter of batch size? Is there any intuition why this problem seems more serious when one uses BALD as the acquisition score? (2) To me, Figure 4 really points out the problem and Figure 5 explains why we want to do AL in a batch manner (due to the computational cost). To avoid confusing though, I would suggest emphasizing that an AL algorithm is considered as good when the "accumulated accuracy" is maximized (Fig 4), as long as it is not at the cost of extra computational burden (Fig 5). Since simply just reading Fig 5 (and the description in the text), one could interpret it as BALD is better since it takes less "time" to achieve 95% accuracy. (3) I'm not sure if I get the meaning of Fig 8. You can probably just randomly permute the RHS and it's going to look equally uniform. Likewise, you can also sort the bins for the BatchBALD's acquisition count, then for sure the difference between the max and min values will be contrasted. Perhaps sorting both of them will be more fair? I would also suggested printing some statistics measuring the dispersion (entropy, range, etc). On a related note, why is it that they end up having different "datasets"? is it that the training set is not exhausted after all of the acquisitions? (4) The section on "scope and limitations" seems a bit hastily written. For example, I don't get the intuition of why BatchBALD is expected to perform well if the test set is balanced, and not so otherwise. Will BALD also perform well if the test set is balanced? Please elaborate. (5) What does the shaded area in all of the figs stand for? "1 stdv from the mean across multiple experiments"? "over test set data points"? (typos) - line110: redundant "p(w" at the end? - line122: I think P_{1:n-1} is of shape c^{n-1} x k ? - line238: "Noisy" estimator

Reviewer 2



Summary: The paper developed an active learning that selects a batch of images that jointly maximizes the mutual information and hence improves the accuracy of the image classifier. This is an extension of Bayesian Active Learning by Disagreement (BALD) acquisition function that computes a mutual information between a set of points and model parameters. This approach outperforms BALD in three datasets: MNIST, RepeteadMNIST and EMNIST. Originality: The paper proposes an interesting idea of selecting samples to annotate jointly by maximizing the mutual information between a batch of samples and model weights instead of selecting them independently for each batch. Basically, the main contribution is the extension of the BALD acquisition function on image data to jointly score samples that would minimize the number of annotations while maximize the performance of the model; thus it extends the mutual information between model prediction and model parameters to between a joint of multiple model predictions. Despite proposing this novel batch selection acquisition function exploiting mutual information between multiple data points the paper should include some related work on batch selection [1,2,3], particularly regarding a discriminative batch model that also exploits the uncertainty of unlabeled data [1], selection of the best samples that match a distribution [2] and exploiting diversity in a mini batch using K-means clustering[3], and include recent literature in active learning [4]. Significance: This paper seems to be a useful contribution to the literature on Batch Mode Active Learning, showing that using the mutual information between a joint set of multiple data points and model predictions outperforms independent selection of samples using BALD and Random selection on three MNIST-based datasets: Repeated MNIST, MNIST and EMNIST, and VarRatios and Mean STD on Repetead MNIST. Moreover, the paper shows that the proposed approach requires less interactions with the oracle (batch size>1) to achieve the same performance when the oracle is asked at each iteration (batch size =1) (upper bound) although the same amount is annotated; hence it reduces the number of times the model is retrained and the number of requests to annotate the data. However, the main weakness of the paper in my view are: (a) the proposed approach should be also tested on more difficult datasets like CIFAR-10, CIFAR-100, CINIC-10 or ImageNet that requires the annotation of large batches in order to have significant improvement on the performance of the model due to the difficulty of the task. Hence, it would demonstrate that usability of the approach on real world problems. Moreover, (b) a comparison with the most current related work on active learning like CoreSet, Variational Adversarial Active Learning, DeepFool Active Learning, or Expected Gradient Length would be interesting to show the need of using batch mode selection. Finally, (c) it would be interesting to add an ablation study where the performance of BatchBALD and BALD is compared with different sizes of the subsampled dataset pool because the performance of BALD depends also a lot on this hyperparameter and it would reduce the chances of having repeated images on RepeatedMNIST. Quality: The idea of having a mutual information between a joint set of multiple data points and parameter and of how it is optimized by applying two approximations (greedy algorithm that selects one sample each time per batch and Montecarlo sampling to avoid exact computation) are certainly good for joint selection of data points while reduces computational time albeit some careless notation mistakes (symbol in Eq. (2), Lemma 2 and line 370 in supplemental material). It would be interesting to show empirically that these two approximations are negligible to the exact solution besides the theoretical proof in Appendix A. Clarity: The paper is clearly written and structured clearly, and the code help me on clarifying an important point in the paper (see below). The section from 3.3 to 3.4.1 could be improved by describing firstly the computation of BatchBALD and then introduce the greedy approximation and the efficient implementation for solving the NP-hard problem. It would be present smoothly the proposed approach and the contributions. However, the paragraphs from line 127 and 131 that describes the MonteCarlo sampling to avoid exact computation should be described how it is computed clearly because it could confuse since given x_1:n-1 fixed the predictions y_1:n-1 are also fixed. Checking the provided code I figured out how it is calculated the P_1:n-1 for the joint mutual information. References: [1] Yuhong Guo and Dale Schuurmans. Discriminative Batch Mode Active Learning. NIPS 2008. [2] Javad Azimi, Alan Fern, Xiaoli Z. Fern, Glencora Borradaile, and Brent Heeringa. 2012. Batch active learning via coordinated matching. ICML12 [3] Fedor Zhdanov. Diverse mini-batch Active Learning. https://arxiv.org/pdf/1901.05954.pdf [4] Samarth Sinha, Sayna Ebrahimi and Trevor Darrell. Variational Adversarial Active Learning. https://arxiv.org/pdf/1904.00370.pdf ------------------------------------------------------------------------------------------------------------------ First of all, I appreciate the author’s effort for addressing all our concerns and improvements, specially adding extra experiments during this short time. Therefore, I upgrade my recommendation as a good submission and voting for his acceptance. Moreover, I would encourage the authors to also include an ablation study of how the subsampling size w.r.t. the unlabeled pool affects BALD and BatchBALD. Based on my experience using BALD, the size of the subsampled samples from the unlabeled pool affects on the final performance of the model; hence I recommend adding this ablation study to improve the quality of the paper.

Reviewer 3



Quality: Pros: Overall, this is a technically sound submission. I really like the proof of the submodularity of the proposed batchBALD acquisition function. Furthermore, the way to estimate that acquisition function using Monte-Carlo as well as the new efficient implementation are quite interesting. Cons: My first concern is about the quality of the estimation in (10) when working with large acquisition sizes $n$. In particular, the number of overall possible configurations of $y_{1:n}$ is $n!$ (# of permutations)--will be extremely large when $n$ increases, while only $m$ samples were chosen. Although this was explained in app. C, it's still unclear for me about the difference between $m$ and $n!$. Furthermore, the experimental results of the submission is not very compelling since they were only conducted on MNIST and its variants. It's would be more convincing if the authors can provide the results on at least one more benchmark data set in the field (e.g., cifar10). Clarity: The submission is clearly written and well organized. However, it's unclear for me about the definition of data repetition. Does a data point x' duplicate the given data point x if x'=x, or these samples are closed enough? That relates to the Alg.1 as well as way to generate the repeated MNIST(sec. 4.1). In particular, Alg. 1 can only guarantee that the new selected data point $x_n$ is different from the previous ones; while in repeated MNIST, a duplicated sample is generated by adding a Gaussian noise to a given sample. Originality: The proposed BatchBALD is a novel extension of one of the most widely studied acquisition function in the Bayesian active learning with disagreement (BALD) framework [10], but targeting the selection of a joint (dependent) batch data samples to improve the data diversity. The paper also introduces the use of a greedy approximation algorithm as well as new ways to estimate the BatchBALD acquisition function. Significance: The main contribution of the paper is to improve the data efficiency of the selected informative data points in BALD w.r.t both the diversity and batch size. The experimental results are quite promising. Post rebuttal comments: I have read the author feedback carefully. Thanks the authors for providing insightful clarifications, especially for providing further experimental results required for the score improvement. Also, based on positive coments/evaluations from fellow reviewers to the paper, I decide to upgrade my score for the paper to 7.

[Author Response · NeurIPS 2019]

We thank the reviewers for taking the time to write these thorough reviews and their appreciation of BatchBALD as a
theoretically grounded way to do batch active learning. We address reviewer 1, 2 and 3 as **R1**, **R2**, **R3**.

**R1-(1)**: Indeed, any acquisition function that does not consider batch dependen-
cies is sensitive to redundant acquisition. Var-Ratios and Mean-Std outperform
random on RMNIST due to being more noisy methods than BALD, which helps
them in this situation, see also lines 222-224 in the paper.
**R1-(2)**: We agree that accuracy is the main metric. We want to highlight that
batch acquisitions significantly reduce the number of times the oracle needs to be
consulted and the model retrained. Figure 4 and 5 show that BatchBALD allows
for batch acquisitions without losing accuracy or incurring extra computation.
**R1-(3)**: We have updated the figure 8 with both sides sorted. The entropy can be
found in figure 7. Training is performed until a suitable accuracy is reached, not until the unlabelled set is empty.
**R1-(4)**: We will reword section 6. Using mutual information encourages acquiring a dataset where the model is
informed equally about each class. For severely unbalanced test sets, the model does not have the right bias and will
under-perform. **R1-(5)**: We use 25%, 75% quartiles for the shaded areas, see line 147 in the paper.
**R1-improvements**: No improvements mentioned.

---

**R2 - Originality**: Thank you for pointing us to additional relevant related work: we have added citations.
**R2 - Significance - a**: EMNIST, with its 47 classes and 112,800 data points, is substantially more difficult for active
learning than regular MNIST. We provide additional results on CINIC-10 (top figure, left). We use 160k training
samples and 20k validation samples, 90k test samples. We use an ImageNet pretrained VGG-16, with a dropout layer
before a 512 HU (instead of 4096) fc layer. We use 50 MC dropout samples, acquisition size 10 and 6 trials. The results
are in the figure above, with the 59% mark reached at 1170 for BatchBALD and 1330 for BALD (median).
**R2 - Significance - b**: We have considered the proposed methods: **CoreSet** uses very large acquisition sizes which is
unrealistic for many applications; **Variational Adversarial Active Learning** is a semi-supervised learning method,
and we consider using BatchBALD in that setting as future work; **DeepFool Active Learning (DFAL), Expected**
**Gradient Length** reported for MNIST that they achieve 83% and 84% (DFAL) and 59% and 38% (EGL) accuracy
after 100 samples with LeNet5 and VGG-8 respectively, while BatchBALD obtains 90% accuracy with only 90 samples
with a smaller model, see table 1, line 196. We will add these baselines to our table.
**R2 - Significance - c**: We perform an ablation study on RepeatedMNIST (top figure, right): we vary repetitions from
0 to 4 (paper reports 2 repetitions, 0 still adds noise to MNIST), same setup as in section 4.1. BatchBALD performs
the same on all repetition numbers (100 data points till 90%). BALD achieves 90% accuracy at 120 data points (0
repetitions), 160 data points (1 repetition), 280 data points (2 repetitions), 300 data points (4 repetitions). This shows
that BALD and BatchBALD behave as expected. We will add the corresponding plot to the paper.
**R2 - Quality**: We understand the concern about the approximations. For any interesting dataset, we can't compute
the joint MI exactly. Therefore, we rely on our mathematical result, with proof in Appendix A., which bounds the
approximation. For Monte Carlo sampling, we have conducted further investigation into the variance of the estimator
and concluded that, using consistent dropout masks, line 152 of the paper, and the used batch sizes, the variance doesn't
affect the order of the top points. We will add the analysis and results to the appendix.
**R2 - Clarity**: We will restructure the methods section. We appreciate the time you took to look at the code. We attempt
to make your point in line 126, where we state we can sample once at the beginning of the algorithm. We will emphasize
that point in lines 127-131 for clarity.
**R2 - Improvements**: We address all requests: we add results for the CINIC-10 (a combination of CIFAR-10 and down-
scaled ImageNet) dataset you proposed, see **R2 - Significance - a**. We added more comparisons in **R2 - Significance -**
**b** and perform an ablation on the repeated MNIST setup in **R2 - Significance - c**.

---

**R3 - cons 1**: Indeed, the quantity in (10) becomes very large as $n$ increases. In fact, the number of configurations $\hat{y}_{1:n}$
is $C^n$. In section 3.4.1 we discuss two methods to reduce the computational load: efficiently caching previous results
(possible due to the greedy approach) and beyond the first 4 acquisitions, we use Monte Carlo sampling with $m$ samples.
In our setup, $m$ is chosen according to our computational budget (10,000), while $n$ is varied per problem.
**R3 - cons 2**: See **R2 - Significance - a** (eval on CINIC-10, a combination of CIFAR-10 and down-scaled ImageNet).
**R3 - clarity**: You are correct in saying the duplicated ("repeated") samples are generated by adding Gaussian noise:
$x' = x + N(0, 0.1I)$. We don't explicitly ignore or remove duplicates. If we acquire a particular point, then its near
duplicates (with added Gaussian noise), will not increase our information about the true parameters, and BatchBALD
will prefer other points instead.
**R3 - improvements** We address all requests: 1. see **R3 - clarity**. 2. see **R3 - cons 1**. 3. Mean and stddev: BatchBALD
- MNIST acq size 10: 90% acc at 91.7 with $\sigma = 16.7$, 95% acc at 211.7 with $\sigma = 27.3$; BALD - MNIST acq size 10:
90% at 148.33 with $\sigma = 29.1$, 95% acc at 273.33 with $\sigma = 26.9$. All results are significant with $\alpha = 0.05$. 4. See **R3 -**
**cons 2**.

[Meta-Review · NeurIPS 2019]

The paper proposes BatchBALD, a batch acquisition function for sample selection in active learning. A greedy optimization algorithm is presented for efficient sample selection and BatchBALD score maximization. The reviewers and AC agree that this is an interesting work and that the approach is clearly presented and convincing. In addition the author response satisfactorily addresses the points raised in the reviews. Hence acceptance is recommended.